# An IoT Real-Time Potable Water Quality Monitoring and Prediction Model Based on Cloud Computing Architecture

**DOI:** 10.3390/s24041180

**Published:** 2024-02-11

**Authors:** Rita Wiryasaputra, Chin-Yin Huang, Yu-Ju Lin, Chao-Tung Yang

**Affiliations:** 1Department of Industrial Engineering and Enterprise Information, Tunghai University, Taichung 407224, Taiwan; rita.wiryasaputra@ukrida.ac.id (R.W.); huangcy@thu.edu.tw (C.-Y.H.); yujulin@thu.edu.tw (Y.-J.L.); 2Informatics Department, Krida Wacana University, Jakarta 11470, Indonesia; 3Department of Computer Science, Tunghai University, Taichung 407224, Taiwan; 4Research Center for Smart Sustainable Circular Economy, Tunghai University, Taichung 407224, Taiwan

**Keywords:** Arduino, binary classifiers, drinking-water, Internet of Things (IoT), machine learning, monitoring, notification

## Abstract

In order to achieve the Sustainable Development Goals (SDG), it is imperative to ensure the safety of drinking water. The characteristics of each drinkable water, encompassing taste, aroma, and appearance, are unique. Inadequate water infrastructure and treatment can affect these features and may also threaten public health. This study utilizes the Internet of Things (IoT) in developing a monitoring system, particularly for water quality, to reduce the risk of contracting diseases. Water quality components data, such as water temperature, alkalinity or acidity, and contaminants, were obtained through a series of linked sensors. An Arduino microcontroller board acquired all the data and the Narrow Band-IoT (NB-IoT) transmitted them to the web server. Due to limited human resources to observe the water quality physically, the monitoring was complemented by real-time notifications alerts via a telephone text messaging application. The water quality data were monitored using Grafana in web mode, and the binary classifiers of machine learning techniques were applied to predict whether the water was drinkable or not based on the data collected, which were stored in a database. The non-decision tree, as well as the decision tree, were evaluated based on the improvements of the artificial intelligence framework. With a ratio of 60% for data training: at 20% for data validation, and 10% for data testing, the performance of the decision tree (DT) model was more prominent in comparison with the Gradient Boosting (GB), Random Forest (RF), Neural Network (NN), and Support Vector Machine (SVM) modeling approaches. Through the monitoring and prediction of results, the authorities can sample the water sources every two weeks.

## 1. Introduction

One of the United Nations’ goals is to provide clean drinking water and improved sanitation to all by 2030. Nevertheless, millions still suffer without safe drinking-water and proper sanitation despite the progress achieved by some countries like Greece, Iceland, Germany, Singapore, and Kuwait [1]. Polio, cholera, typhoid, dysentery, and diarrhea are transmitted through poor water quality and improper sanitation. Drinking water safety is essential because diarrhea has a mortality rate of about 829,000 people per year [2]. Inadequate water-treatment plans and inadequate water-distribution networks lead to low-quality water, which is prone to pollution [3]. Based on the World Health Organization’s (WHO) drinking water guidelines, consumer acceptability is defined by visual, taste, and smell characteristics. Electrical conductivity (EC), temperature, and turbidity are used to measure physical water quality; meanwhile, chemical parameters such as dissolved oxygen (DO) and pH indicate chemicals that represent risks in drinking water samples [4]. IoT and machine learning are two emerging technologies that allow cross-domain researchers to build various systems that benefit human life. An IoT real-time package adopts different devices on which data can be transmitted and passed without human intervention. The IoT system provides several benefits over conventional techniques, including reduced costs, efficient travel, and data collection, adaptability over different system phases, and the ability to predict ideal values for unreachable locations [5]. Researchers use parameters such as DO, pH, temperature, EC, and turbidity for water quality monitoring. Researchers use a minimum of two (2) sensors for the corresponding elements in the water monitoring systems including pH sensors and TDS sensors or water temperature sensors [6,7,8,9,10,11]. Ultrasonic sensors can detect water quality parameters, integrating it with the WeMos D1 mini through its built-in Wi-Fi capabilities [7]. An Arduino platform has an integrated development environment which makes it easy for users to develop their projects. Thanks to its simplicity and accessibility, most researchers have combined it with the sensors [6,8,9,10]. Moreover, utilizing the ThingSpeak IoT platform can also enhance the monitoring system’s features [7,9,12]. The monitoring system’s complexity can be extended to prediction systems that employ the collected data. In terms of implementation, machine learning methods involve predicting data with variable parameters. Researchers have used one robust machine learning algorithm—the SVM model—to solve medical classification problems [13,14], and environmental issues [3]. Aldhyani revealed the SVM was a profound model for water quality classification forecasting compared to the Naive Bayes approach and the K-nearest neighbor approach. It had the highest percentage accuracy at 97.01 [15]. However, in a different experiment conducted by Nasir, the CATBoost model had the highest percentage accuracy, of 94.51, rather than the SVM model. The performance of SVM model achieved an accuracy 80.68%, which is included in the lowest category compared to the performances of RF, DT, CATBoost, MLP, XGBoost [16]. The GB model is another well-recognized machine learning classification method [17,18]. Khan proposed a water quality prediction model based on the weighted arithmetic index method and principal component analysis (PCA). Compared to the LR, GB, RF, and the SVM, the GB model’s performance reached a perfect level of classifying the water quality status [5]. The rules used by a DT model are easy to understand and can directly process the features in numerical and categorical forms [19,20,21]. The SVM and GB classifiers are potential solutions to regression and classification issues. Using the simulation of a water for drinking and irrigation dataset, Ajayi revealed that, between three machine learning approaches, namely, RF classifier, Logistic Regression (LR), and SVM, LR performed better for drinking water classification purposes than SVM [22]. Table 1 presents the prediction models used to support water monitoring systems, where the most popular approach is SVM. Our research has two goals. One is to come up with a low-cost potable water monitoring system. A web-based tool and real-time notifications via the LINE social media application are used as the mobile messenger for monitoring drinking-water quality in real-time. The second goal is to compare the forecasting scheme between the decision and non-decision trees using sensor data so the authorities can monitor, regulate, and check the drinking-water supply every few months. The proposed research is structured as follows: Section 1 and Section 2 provide the research background, material, and prediction methods that are used, respectively. In Section 3, the experiment results are presented. Section 4 discusses the experiment carried out with the proposed system. The last section consists of the conclusions and future works.

## 2. Materials and Methods

A schema of the proposed drinking-water quality monitoring method is represented in Figure 1.

### 2.1. IoT Sensor Packages for Water Quality Monitoring

There are numerous built-in water quality detectors at various prices on the market; however, not all are compatible with experimental conditions. A monitoring system can also be achieved with self-assembly low-cost sensors. The IoT sensor packages in the proposed water quality monitoring comprise a Total Dissolved Solids (TDS) sensor, pH sensor, DO sensor, and temperature sensor, and are shown in Figure 2.

#### 2.1.1. TDS Sensor

TDS sensors come in a PPM format that consists of three pins: data, VCC, and ground. Electrical conductivity helps to define the quantity of impurities in terms of inorganic salts (salinity) and contaminants like iron, zinc, or aluminium. In the experiments, there was no guarantee that the sensor measurements that were calibrated would not experience errors. When the domestic water data were periodically sent through a cloud Wi-Fi connection, the TDS measurement had an error with a percentage of 0.62 and failed at pH 0.94 [6,23].

#### 2.1.2. pH Sensor

When assessing acidity and/or alkalinity, the potential of Hydrogen (pH) can be used, with the pH scale ranging from 0 to 14 and neutral being 7. A value of more than 7 means basic or alkaline. On the contrary, a value of less than 7 is an acid. According to WHO standards, the safe pH range of drinking water is 6.5–8.5 with higher pH being favorable for neutralizing stomach acid. Detecting hydrogen ion concentration with a pH sensor is performed using three pins—output, power, and ground—where the output pin is connected to the microcontroller’s analog input pin. A pH sensor has a sensor electrode and a reference electrode. The sensor electrode serves as a measurable indicator of the hydrogen ion concentration which is related to the pH value. The mechanism used is that the current flowing through the sensor electrodes is directly proportional to the concentration of hydrogen ions. When the concentration of hydrogen ions increases, the current flowing through the electrodes has the same experience. In contrast, for the pH value, as the concentration of hydrogen ions increases, the pH value decreases. In particular, the nature of the content of the average indicator value is typically dictated by the electrochemical differential potential associated with a reference solution within the respective electrode system whilst the unknown solution is outside of it [24].

#### 2.1.3. DO Sensor

DO is one of many important water quality indicators, referring to free and non-compound oxygen in water. The DO concentration indicates the environment’s ability to regulate itself; a high DO level reflects the fast speed of various pollutants in the water which occur during degradation. On the other hand, contaminants in water degrade more slowly when the DO level is low. Generally, having a high DO concentration in drinking water is not ideal for human health. Even though it does not threaten human health directly, it influences the water quality indirectly. Very high concentrations of DO can also cause corrosion on steel pipelines. It is more relevant in aquatic ecosystems to support the aquatic organisms’ life. The researchers [25,26] provided an appropriate DO level as a prerequisite for aquatic organisms’ lives by integrating the prediction module along with a DO sensor to avoid low DO levels.

#### 2.1.4. Temperature Sensor

The right temperature should be maintained because higher temperatures enable bacteria to multiply faster in water than room temperature conditions. A direct proportion exists between the output voltage of waterproof digital temperature sensors and temperature variations from a minimum of minus 55 °C to a maximum of 125 °C [7,27].

#### 2.1.5. NB-IoT

The equipment on the sensor is also equipped with communication technology related to information retrieval. The transmission of data from end devices to the web server can overcome distances of more than 2 km with the aid of a low-power wide-area network (LPWAN) infrastructure. The major LPWAN technologies include LoRa and NB-IoT which boasts low power consumption and optimal accessibility for many devices. NB-IoT enhances the network simplicity as it is stable and reliable; thus, its simple design enables it to coexist with the existing LTE wireless networks, addressing the demands of cellular networks [28,29].

### 2.2. Machine Learning Models

The proposed potable water monitoring uses machine learning models to generate predictions for drinking-water quality; Figure 3 shows the workflow of predictive drinking water monitoring.

#### 2.2.1. DT

A DT is a supervised learning classifier used as a learning base in the GB algorithm, and every new model has been built according to which direction the gradient slope indicated an error concerning the previous model that had been used. DT has several characteristics: superficiality, higher performance accuracy while using less than a thousand data points, and correctly classifying all testing samples [5]. Partitioning the sample dataset using the best samples with high entropies results in the collection of decision rules through a decision tree approach.

#### 2.2.2. SVM

The use of SVM provides better performance on high dimensional data, good quality data, and as a discriminant analysis classifying different classes using a hyperplane, lowering the error rate when its output is very low when dealing with overlapped classes. Even though the SVM is robust to noise [30], it is important to note that SVM has issues with parameter selection, computational time, sizes, and unbalanced datasets [14]. An alternative strategy for overcoming SVM’s weakness is selecting the right type of kernel function [16].

#### 2.2.3. RF

The RF model, as the supporting tool for the classification process, combines bootstrap aggregation and randomization. The model is built from several decision trees, and thereby, there is no relationship between each tree. In classification problems, every new data point will be labeled by each decision tree and has an output. The key characteristic of RF is its adaptability to datasets with many features even though there are scattered data or continuous data. The RF model processes high-dimensional data directly and without normalization. RF has two role learners—a weak learner and a strong learner. It uses the classification and regression decision tree (CART) as a weak learner to select all the features and combines multiple weak learners to construct a strong learner.

#### 2.2.4. Evaluation Model

There are four elements (TP, TN, FP, FN) in the evaluation model. TP (True Positive) means an actual sample is identified correctly. TN (True Negative) happens after correctly predicting successful negative samples. FP (False Positive) is a misprediction of a positive sample, whereas FN (False Negative) draws mispredictions of negative samples (or positive samples that were not predicted). Equations (Equation 1)–(Equation 4) refer to Accuracy, F1-score, and Receiver operating curve (ROC), respectively, and were used to evaluate the model’s performance. Accuracy refers to how well the predicted output aligns with the actual output by calculating the total number of anticipated values and dividing this by all predictions made by the model. The F1-score or harmonic mean is a statistical measure used for computing the average of precision and recall and to estimate model performance accuracy. The recall is how sensitive the model is to the proportion of positive results or drinkable water. Specificity is the percentage of undrinkable water. The ROC curve indicates that the model is better if it is close to the ideal node value of 1.
(1)$$Accuracy=\;\frac{TP+TN}{TP+TN+FP+FN}$$
(2)$$Recall=Sensitivity=\;\frac{TP}{TP+FN}$$
(3)$$Precision=\;\frac{TP}{TP+FP}$$
(4)$$F1-score=\;\frac{2*Precision*Recall}{Precision+Recall}$$

## 3. Results

The water dataset in the experiment was collected randomly from the Tunghai campus’ drinking fountains and tap water, which follows a set of IoT gears on an NB-IoT with a gateway in a 5G network. Users can monitor the water condition via web mode, and the water quality indicators are illustrated in Figure 4. Not only can the user access a browser to monitor the water condition, but they also receive notification messages via their mobile phone. Figure 5 shows the outcome of the LINE social media notifications in which the content involves the parametric value followed by its units of measurement and the status indicator. The experiment had an average unit temperature of 25.62 °C, whereas the temperature of most non-drinkable water is less than 25 °C. Figure 6 shows the distribution values compared to the temperature. The division of the data allocates 60% for training the model, 20% for validating the model, and 10% for data testing. All the prediction models were evaluated based on accuracy and the ROC; the results are presented in Figure 7, Figure 8, Figure 9, Figure 10 and Figure 11, respectively. Figure 11 is a visual representation of the evaluations obtained in the test, train, and validate segment for the DT classifier. The values for the performance of the SVM model’s evaluation are shown in Table 2. In the evaluation stage, using validation data, the SVM model gained 97.53 percent for its accuracy metric. This high accuracy level indicates the model’s ability to correctly classify the instances within the dataset. Meanwhile, the error tolerance level did not exceed 0.05. Additionally, it highlights the reliability of minimizing misclassifications. From the perspective of competence of the model to the predict drinkable water, the proportion of SVM model’s positive results reached 99.78%. In terms of assessing water quality, the SVM model is capable of distinguishing between drinkable and non-drinkable water samples. Table 3 shows a comparison of the evaluation machine learning models in proposed systems. The RF, GB, and SVM had balanced accuracy and F1-score values of 0.9775 and 0.977, respectively, where the performance of NN was the lowest among them. The balanced sensitivity and specificity, demonstrated by the RF, GB, and SVM, indicate that the models had robustness in the classification tasks. The DT classifier performs outstandingly, achieving the highest percentage among the evaluated models, including RF, GB, SVM, and NN. The DT yielded an accuracy in predicting the values of the pH parameters with an approximate percentage of 97% with an observed EC value of 0.001, TDS value of 0.03125, and TD value of 0.963 according to Figure 12. The DT model can accurately predict water quality parameters.

## 4. Discussion

A detailed discussion of the proposed potable water quality monitoring system is given in this section. In order to create a friendly atmosphere, a private university in Taiwan, Tunghai University, provides 249 drinking fountains for its academics and visitors. The drinking fountains are checked every three months.The drinking fountains are placed outdoors, and one is also available on each floor of every building; the distance between buildings is approximately 500 m. Referring to the system design in Figure 1, the first layer is the sensing layer. In the sensing layer, the reliable information is obtained by the collecting data from a variety of sensors that check different water parameters for drinkability. Sensors were chosen based on ease of use, economical factors, and parameter measurability. All the sensors needed to be calibrated before they were used. This included the calibration of the DO sensor which used sodium hydroxide with a concentration of 0.5 mol/L; an indicator of the pH sensor having already been calibrated was the output value of the voltage pulse, which was approximately three. A series of linked sensors is connected to an Arduino microcontroller board. An Arduino microcontroller board becomes a data processing unit and a server, and all of the data sensors’ acquisition goes on it. The Arduino microcontroller is a low-end processing device that lays just at the edge of the sensing layer for gathering, aggregating, and filtering data based on the program, which is written in C language. Lightweight communication protocols and power are essential resources in a water monitoring network. Efficient operation is crucial for LPWAN devices and should be operational for extended timeframes in autonomous IoT systems. A continuous data stream is transmitted by the physical devices in the IoT via the NB-IoT module. The Hypertext Transfer Protocol (HTTP) acted as a request response provided that there was connectivity between the client and the cloud server and enabled communication with all IoT devices. The Python script is used to connect the Arduino to the server. The script contains the accessing token number for the mobile messenger and threshold indicators for labeling and redirection instructions so that data can be stored in the database. There remains a risk that errors may occur in the calibrated sensor measurements. Therefore, the collected data comprised 2837 samples and excluded the null values. All sensor readings data are stored in the relational database management system known as MariaDB MySQL. The Grafana application, as a multi-platform open-source visualization web application, is connected to the database, and the water quality information is presented as graphics so the end-users can monitor it easily. The water quality dataset was organized with two label categories: a 0 value means drinkable while a 1 value signifies non-drinkable. The prediction model is also incorporated in this study to enhance the performance of a modern potable water monitoring system and was provided as a service in the cloud layer. Machine learning algorithms make machines intelligible with the ability to discern complicated data by self-learning. A specific prediction was made on vast amounts of stored data. As a comparative study, the RF, GB, Neural Network (NN), SVM, and DT approaches were used to classify water as drinkable or non-drinkable. As part of the prediction management stage, NN employed a fully connected neural network model. Its architecture comprises eighteen input nodes, fifty hidden nodes, and one hidden layer. There were two node outputs when the NN processed the Tanh activation function. Another setting was used in the SVM model, which used a maximum of 25 iterations with a polynomial kernel. Using 100 trees and a maximum of two branches in the tree cohorts, the RF used a maximum of 20 depth levels. While GB had a learning rate of 0.1 and Gaussian distribution, the range of tree nodes was nine to eleven, with a maximum of four levels of depth, and a maximum leaf size of 367. To achieve a good performance in the models and avoid the overfitting issues that may occur because of the robust dataset, preprocessing with feature extraction technique was carried out. As a statistical analysis, PCA transforms a dataset’s feature into a new set of uncorrelated variables known as principal components while retaining the most dominant water quality parameters. When calculating eigenvalues and eigenvectors, the PCA uses a correlation matrix. The data were collected by subtracting the mean and dividing it by the standard deviation for each variable. The result of the correlation matrix computation using the standardized data was the pairwise correlations between all pairs of variables in the dataset. The correlation coefficient measures the strength and direction of a linear relationship between variables. After the direction and strength were known, the eigenvalue, which was sorted in descending order, was broken down on the correlation matrix. Each eigenvector represents a principal component, and its associated eigenvalue indicates the amount of variance explained by the component. Principal components associated with higher eigenvalues explain more variance in the data and a linear combination of the original variables. The original data can be projected onto the selected principal components to obtain a reduced-dimensional dataset representation. The maximum number of principal components to pass as successor nodes was set at 20. The cumulative proportional variance cutoff value is in 0.99 with a minimum variance increment of 0.001. The Pearson correlation analysis, as a normalization method, helps make variables dimensionless or give them similar distributions. Standard deviation measures the consistency of different datasets that are spread out from their average value. Figure 6 shows that water temperature remained constant during the period of evaluation and its standard deviation value was, in fact, very far away from the mean. This is contrary to how water purity should be measured, where the mean value of TDS was 43.3 and the standard deviation was almost double, thus indicating that those values fluctuate. The selection found that pH was the most dominant water quality parameter. Although the SVM model is commonly used in water quality research, its performance was not better than DT in this study. The higher the evaluator’s F1-score, which combines precision and recall measurements, the better and more significant the model performance. Based on Table 3, DT, which worked in a max tree depth of ten, attained the highest F1-score compared to all the others in this study and became the best-recommended classification among the others. It is in contrast to the NN. The F1-score is influenced by the cutoff value, which ranges from 0 to 1 in increments of 0.05. With a cutoff of 0.5, the F1-score achieved by the NN model was 0.865 in the test partition, 0.858 in the train partition, and 0.852 in the validate partition. Meanwhile, the RF, GB, and SVM had the same experience in the cohort. The mean squared error indicates how far the model’s estimates differ from the real values found in the dataset. Due to the lowest value for the lowest average squared error, which was 0.0007, the result of DT prediction in Figure 12 corresponds to being 17.14% higher than the observed average pH of 0.82. As the selected model, the DT indicated that most observations are expected to have a lower pH than usual. The next layer after the cloud layer is the application layer. The application layer contains specific software packages that are required for monitoring water parameters via mobiles or websites. Through any website browser like Google Chrome, users can monitor the result of data gathering on water quality from the sensing layer. Through the LINE API, water quality conditions are transmitted as text messages using the LINE application. The notifications on the cellphone display the normal or abnormal status and the water quality value. The indicator status will be displayed as normal if the pH value is 6–8.5 and the TDS level is less than or equal to 500 ppm.

## 5. Conclusions

During an era of industrialization, real-time water quality monitoring is necessary to ensure there is enough purified water for consumption. A self-assembly low-cost sensor with A low power consumption and lightweight transmission performs well in the potable water monitoring system. Remote users can test the drinkable water supply monthly and effectively by using mobile messengers or online browsers. By merging a machine learning approach and water quality data, predictions about the current potable quality status can be made. Compared to SVM, RF, NN, and GB, the DT shows an improved performance in determining the drinkable water quality. For future work, it is possible to enhance the in situ real-time environmental monitoring, particularly within cloud-based enterprise service platforms, such as THE Amazon AWS services, to detect E. coli, chlorine, and harmful algae blooms (HABs). Additionally, the enhanced sensors can be integrated and compared with other prediction models to strengthen the overall monitoring system.

## Figures and Tables

**Figure 1 sensors-24-01180-f001:**
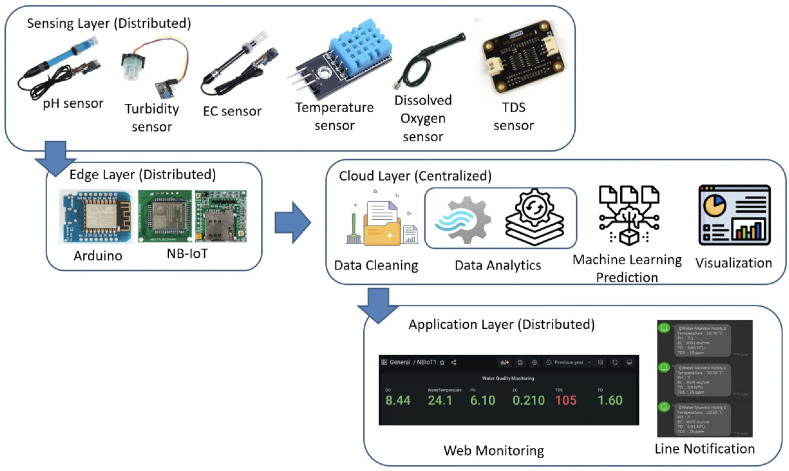
Architecture of the drinking water monitoring system.

**Figure 2 sensors-24-01180-f002:**
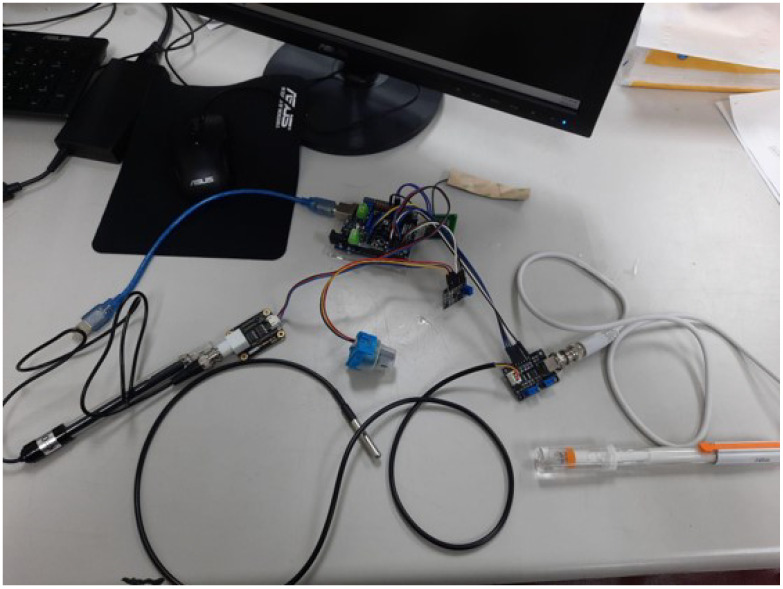
Assembly of sensors.

**Figure 3 sensors-24-01180-f003:**
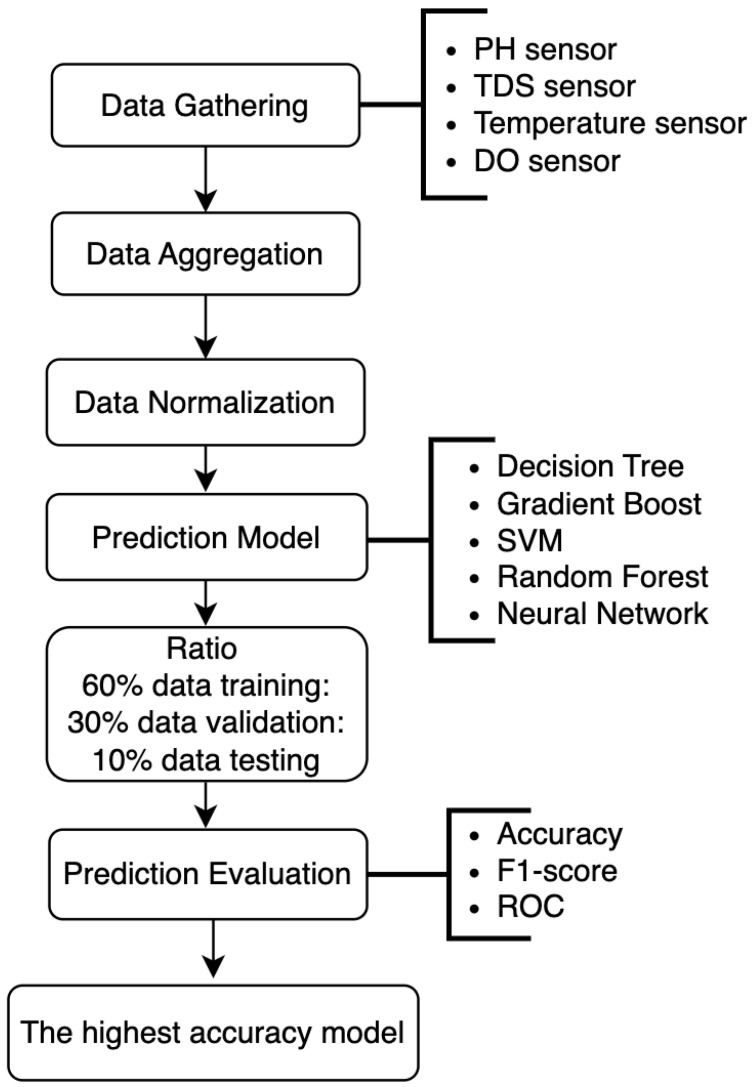
Predictive drinking water monitoring workflow schema.

**Figure 4 sensors-24-01180-f004:**
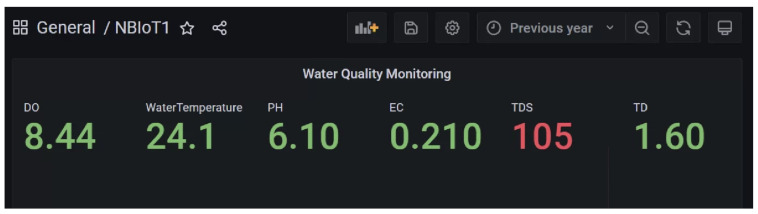
Data monitoring visualization with Grafana.

**Figure 5 sensors-24-01180-f005:**
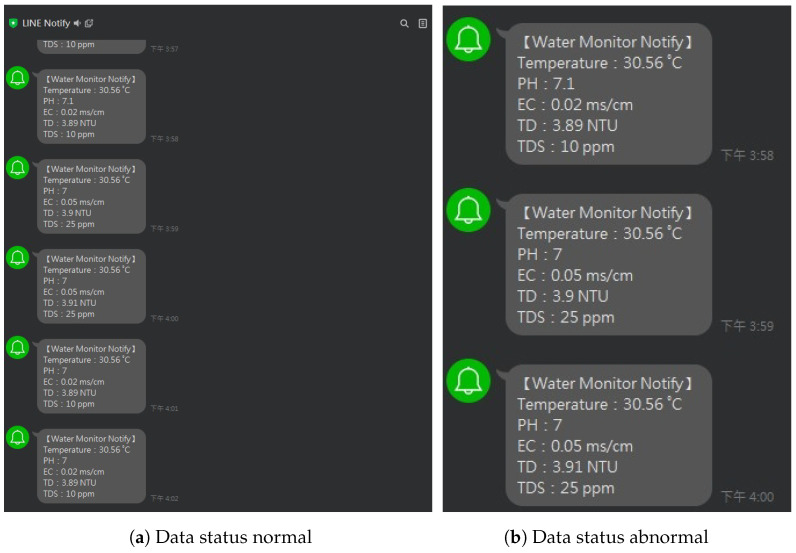
Mobile monitoring with LINE notification.

**Figure 6 sensors-24-01180-f006:**
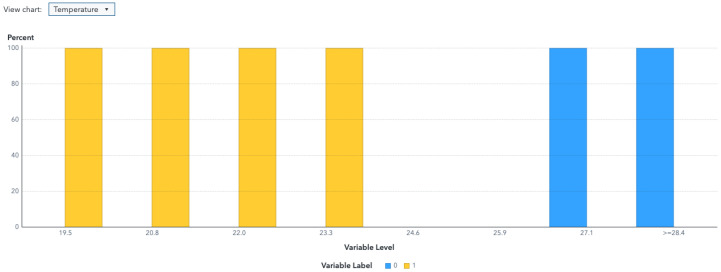
Temperature classification.

**Figure 7 sensors-24-01180-f007:**
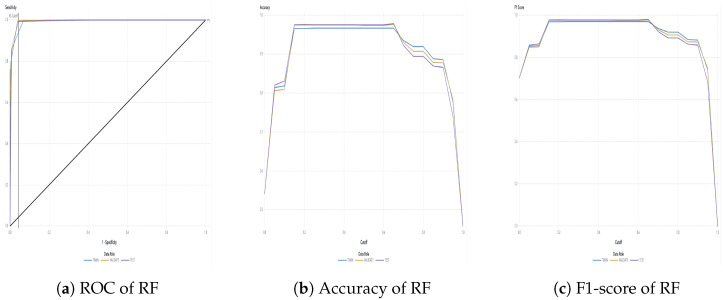
RF classifier evaluation.

**Figure 8 sensors-24-01180-f008:**
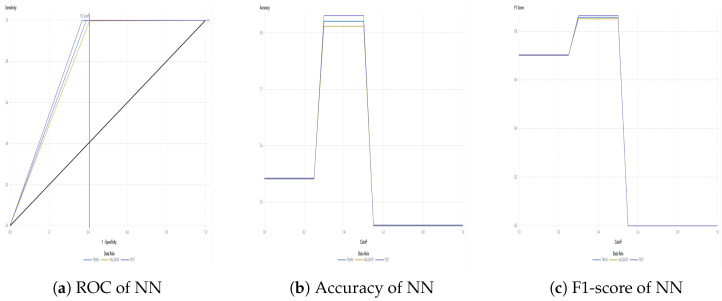
NN classifier evaluation.

**Figure 9 sensors-24-01180-f009:**
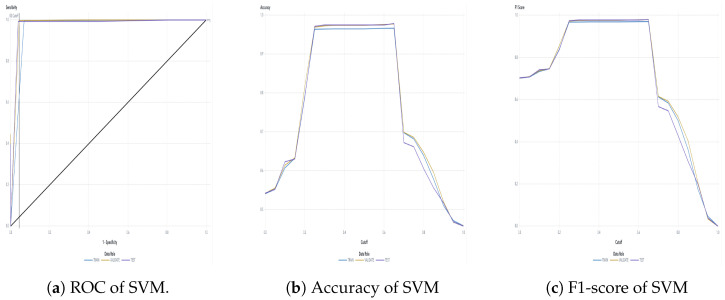
SVM classifier evaluation.

**Figure 10 sensors-24-01180-f010:**
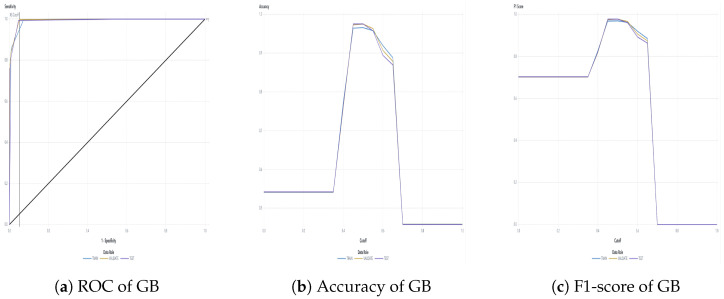
GB classifier evaluation.

**Figure 11 sensors-24-01180-f011:**
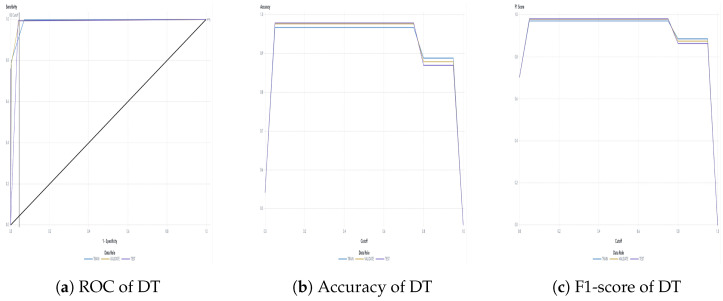
DT classifier evaluation.

**Figure 12 sensors-24-01180-f012:**
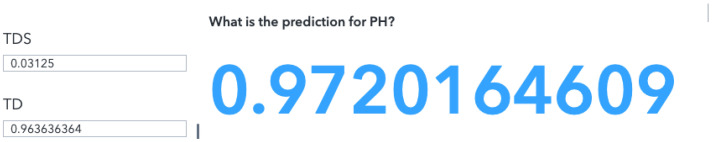
Result of DT classifier.

**Table 1 sensors-24-01180-t001:** Related works with prediction models.

Authors	Prediction Approach	Results
Aldhyani [15]	SVM, Naive Bayes, K-nearest neighbor	SVM had the highest percentage accuracy of 97.01
Nasir [16]	SVM, LR, RF, DT, CATBoost, MLP, XGBoost	CATBoost had the highest percentage accuracy of 94.51
Khan [5]	LR, GB, RF, SVM	GB had the highest percentage accuracy of 100

**Table 2 sensors-24-01180-t002:** SVM evaluation.

Parameter	Training	Validation	Testing
Accuracy	0.9665	0.9753	0.9754
Error	0.0335	0.0247	0.0246
Sensitivity	0.9978	0.9978	0.9935
Specificity	0.9294	0.9486	0.9538

**Table 3 sensors-24-01180-t003:** Comparison machine learning algorithm.

Parameter	DT	RF	GB	SVM	NN
Accuracy	0.97887	0.97535	0.97535	0.97535	0.83099
F1-Score	0.98077	0.97764	0.97764	0.97764	0.8652
ROC	0.99213	0.99505	0.99303	0.98442	0.81538

## Data Availability

Dataset available on request from the authors.

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
