# Peer review of "An IoT Real-Time Potable Water Quality Monitoring and Prediction Model Based on Cloud Computing Architecture"

_sensors, 2024, doi:10.3390/s24041180_

Round 1

Reviewer 1 Report (Previous Reviewer 2)

Comments and Suggestions for Authors

Although the authors propose a relatively new approach to advance water quality monitoring using IoT sensor technology and other emerging platforms for public safety, the present form of the manuscript is not acceptable yet due to the following reason. The revised manuscript is in a better shape than before, but still it is required to be improved. Again, I strongly recommend that someone should proofread thoroughly by considering scientific merits and technical context. The manuscript can contribute to the field with two key aspects. First, LoRa technology can be useful to transmit in-situ data to research office through long-distance communication protocol. Second, the prediction ability of the proposed methods to protect public health for broader impacts.  Nonetheless, the authors barely describe and/or justify these values in the present form. For example, no detailed info about the drinking fountains (indoor or outdoor?, communication range? other interferences?, etc) at the university can help convince the reviwer to adopt your technology. Additionally, the reviewer feels that the text used in the manuscript was infused by “Google Translator” based on the authors’ native language. Perhaps, the authors hire a professional editor to work on the manuscript to improve readability. 

Comments on the Quality of English Language

Line 5-6: Rewrite this sentence, perhaps “Water quality components, such as water temperature, alkalinity…..the Arduino using a low power energy consumption”

Line 7: What do you mean by “the limited resources”?  

Line 22: Outshined à outperformed?

Line 25, WHO. Spell out when it appears for the first time.

Perhaps, Table 1 can be removed from the manuscript. Instead, the authors add the relevant references to the text in line 12.

Line 39. More explanation is necessary for the term “velocity”, What is the velocity in the context of prediction models? Is it an environmental variable? It is not clear.

Line 48. What do you mean by “LINE”? Is it an abbreviation? If so, please spell out when it appears for the first time.

Line 51. “Regularly” How often? In few seconds? In few hours? Or daily? Please specify~~~

Line 55. “the paper is summarized”?

Line 60. “Vary in price” à “various prices”?

Line 63. Do you think the term “IoT ecosystems” is the right term in this manuscript? Perhaps “IoT sensor packages” would be a better term, I guess.

Line 64 – 66: Rewrite this sentence. Fig 2.????

Author Response

Reviewer1

Reviewers’ comment:

Line 5-6: Rewrite this sentence, perhaps “Water quality components, such as water temperature, alkalinity…..the Arduino using a low power energy consumption”

Response:

  • Thank you for pointing this out.
  • Authors have already revised the sentence in lines 5-7 : Water quality components data, such as water temperature, alkalinity or acidity, and contaminants, were obtained through a series of linked sensors.

Reviewers’ comment:

Line 7: What do you mean by “the limited resources”?  

Response:

  • We thank the reviewer for pointing this out.
  • Refer to the word “the limited resources” , what is meant is the limited human resources. We have revised the sentence in lines 8-10: Due to limited human resources to observe the water quality physically, the monitoring is complemented by real-time notifications alerts via a telephone text messaging application.

Reviewers’ comment:

Line 22: Outshined à outperformed?

Response

  • We greatly appreciate the feedback from the reviewer.
  • We have revised the sentence in lines 15-17: the performance of the Decision Tree (DT) model was more prominent in comparison with the Gradient Boosting (GB), Random Forest (RF), Neural Network (NN), and Support Vector Machine (SVM) modelling approaches.

Reviewers’ comment:

Line 25, WHO. Spell out when it appears for the first time.

Response

  • We greatly appreciate the feedback from the reviewer.
  • The authors have revised the abbreviation in line 29.

Reviewers’ comment:

Perhaps, Table 1 can be removed from the manuscript. Instead, the authors add the relevant references to the text in line 12.

Response

  • We greatly appreciate the feedback from the reviewer.
  • We have already removed Table 1 from the manuscript and made the sentences with the relevant references in lines 40-47.

Reviewers’ comment:

Line 39. More explanation is necessary for the term “velocity”, What is the velocity in the context of prediction models? Is it an environmental variable? It is not clear.

Response

  • We greatly appreciate the feedback from the reviewer.
  • The authors have changed the sentence and do not use term “velocity”. The sentence becomes: The rules inferred by a DT model are easy to understand and can directly process the features in numerical and categorical forms (in lines 62-63).

Reviewers’ comment:

Line 48. What do you mean by “LINE”? Is it an abbreviation? If so, please spell out when it appears for the first time.

Response

  • We greatly appreciate the feedback from the reviewer.
  • The authors have already specified the word “LINE” with LINE social media application (in line 70). The LINE social media application like Whatsapp, Instagram, and Facebook.

Reviewers’ comment:

Line 51. “Regularly” How often? In few seconds? In few hours? Or daily? Please specify~~~

Response

  • We greatly appreciate the feedback from the reviewer.
  • We have specified the word “regularly” with every few months (in line 74).

Reviewers’ comment:

Line 55. “the paper is summarized”?

Response

  • We greatly appreciate the feedback from the reviewer.
  • We have revised the sentence, and it becomes: The last section consists of the conclusions and future works (line 77).

Reviewers’ comment:

Line 60. “Vary in price” à “various prices”?

Response

  • We greatly appreciate the feedback from the reviewer.
  • We have already revised the sentence line 83: There are numerous built-in water quality detectors at various prices in the marketplace

Reviewers’ comment:

Line 63. Do you think the term “IoT ecosystems” is the right term in this manuscript? Perhaps “IoT sensor packages” would be a better term, I guess.

Response

  • We greatly appreciate the feedback from the reviewer.
  • We have already revised the sentence in line 85: IoT sensor packages of the proposed water quality monitoring comprising TDS sensor, pH sensor, DO sensor, and temperature sensor, and are shown in Fig.2.

Reviewers’ comment:

Line 64 – 66: Rewrite this sentence. Fig 2.????

Response

  • We greatly appreciate the feedback from the reviewer.
  • We have already eliminated the redundant sentence, and Fig.2. is linked to the IoT sensor packages in sentence lines 85-87: IoT sensor packages of the proposed water quality monitoring comprising TDS sensor, pH sensor, DO sensor, and temperature sensor, and are shown in Fig.2.

Reviewer 2 Report (Previous Reviewer 3)

Comments and Suggestions for Authors

Authors have adressed requested change. So, The paper can be accepted in current forme.

Author Response

Reviewers’ comment:

The methods and conclusion are described 

Response

  • We greatly appreciate the feedback from the reviewer.
  • We have already added more descriptions on methods (line 223-227, 261-276) and conclusions (311-314) with purple color highlight.

Round 2

Reviewer 1 Report (Previous Reviewer 2)

Comments and Suggestions for Authors

Thanks for revising the manuscript. 

The current version of the manuscript is improved significantly. Now, the reviewer will accept this manuscript with the minor revision option. 

1. Line 17-18, Through the "hourly" or "Daily" monitoring and prediction of results....Please specify monitoring time step. For example, every 15- second? every 5-minute? hourly? or daily? Although the authorities can make a decision on their own time step, such as "monthly" noted in the manuscript, the reader may want to know the data collecting time step using the proposed sensing method. 

2. Line 35. Perhaps, "An IoT real-time sensor package" rather than "An IoT real-time ecosystem" is better expression in the overall context of the project. 

3. Line 42. Please consider adding the following reference in references [6 - 10] because IoT sensor devices were used for real-time water quality monitoring activities. 

Ryu, J. 2022. "UAS-based real-time water quality monitoring, sampling, and visualization platform (UASWQP)", Hardware X 11, April 2022 at: UAS-based real-time water quality monitoring, sampling, and visualization platform (UASWQP) - ScienceDirect

Line 46. Additionally, consider adding the following reference in the ThingSpeak IoT platform references [7, 9] because the cloud-based data sharing platform (e.g., ThingSpeak) was used for water quality monitoring activities. 

Ryu, 2022. "Prototyping a low-cost open-source autonomous unmanned surface vehicle for real time water quality monitoring and visualization", HardwareX, October, 2022, doi: 10.1016/j.ohx.2022.e00369

Line 319. Perhaps, the authors want to rephrase the last sentence to: 

"For future work, various sensors can be better equipped to detect E. coli, chlorine, and harmful algae blooms (HABs) to enhance the in-situ real-time environmental monitoring at cloud-based enterprise service platform, such as Amazon AWS services for broader impacts"

Comments on the Quality of English Language

The revised manuscript has been improved significantly so the reviewer is happy to accept the manuscript with a minor revision. 

Author Response

Reviewers’ comment:

Line 17-18, Through the "hourly" or "Daily" monitoring and prediction of results....Please specify monitoring time step. For example, every 15- second? every 5-minute? hourly? or daily? Although the authorities can make a decision on their own time step, such as "monthly" noted in the manuscript, the reader may want to know the data collecting time step using the proposed sensing method. 

Response

  • We greatly appreciate the feedback from the reviewer.
  • We have already revised the monitoring time in lines 17-18: Through the monitoring and prediction of results, the authorities can sample the water sources every two weeks.

Reviewers’ comment:

Line 35. Perhaps, "An IoT real-time sensor package" rather than "An IoT real-time ecosystem" is better expression in the overall context of the project. 

Response

  • We greatly appreciate the feedback from the reviewer.
  • We have already revised the sentence in lines 35: An IoT real-time package adopts different devices on which data can be transmitted and passed without human intervention.

Reviewers’ comment:

Line 42. Please consider adding the following reference in references [6 - 10] because IoT sensor devices were used for real-time water quality monitoring activities. 

Ryu, J. 2022. "UAS-based real-time water quality monitoring, sampling, and visualization platform (UASWQP)", Hardware X 11, April 2022 at: UAS-based real-time water quality monitoring, sampling, and visualization platform (UASWQP) – ScienceDirect

Response:

  • We greatly appreciate the feedback from the reviewer.
  • We have already revised in lines 40-42, and added the citation with the reference in line 358: Researchers used a minimum of two (2) sensors to the corresponding elements on the water monitoring systems-those were the pH sensor and TDS sensor or water temperature sensor [6-11].

Reviewers’ comment:

Line 46. Additionally, consider adding the following reference in the ThingSpeak IoT platform references [7, 9] because the cloud-based data sharing platform (e.g., ThingSpeak) was used for water quality monitoring activities. 

Ryu, 2022. "Prototyping a low-cost open-source autonomous unmanned surface vehicle for real time water quality monitoring and visualization", HardwareX, October, 2022, doi: 10.1016/j.ohx.2022.e00369

Response:

  • We greatly appreciate the feedback from the reviewer.
  • We have already revised in lines 46-47, and added the citation with the reference in line 360: Moreover, utilizing the ThingSpeak IoT platform can also enhance the monitoring system features [7,9,12].

Reviewers’ comment:

Line 319. Perhaps, the authors want to rephrase the last sentence to: "For future work, various sensors can be better equipped to detect E. coli, chlorine, and harmful algae blooms (HABs) to enhance the in-situ real-time environmental monitoring at cloud-based enterprise service platform, such as Amazon AWS services for broader impacts"

Response:

  • We greatly appreciate the feedback from the reviewer.
  • We have already revised in lines 317-321: For future work, it is possible to enhance the in-situ real-time environmental monitoring, particularly within cloud-based enterprise service platforms, such as Amazon AWS services, to detect E. coli, chlorine, and harmful algae blooms (HABs). Additionally, the enhanced sensors can be integrated and compared with other prediction models to strengthen the overall monitoring system.

This manuscript is a resubmission of an earlier submission. The following is a list of the peer review reports and author responses from that submission.

Round 1

Reviewer 1 Report

Comments and Suggestions for Authors

The amount of data being shown is very small. Therefore, it's unclear what the motivation of the research is and what conclusions were derived.

*The authors improved a real-time water quality monitoring system.
I think this paper relevant in the field and address a specific gap in the field.The article proposed that the Decision Tree shows an improved
performance in determining the drinkable water quality.

*The conclusions are not consistent with the evidence and arguments presented and they don't address the main question posed.The authors mentioned that the first objective is coming up with a low-cost potable water monitoring system. However, nothing about them was shown and discussed. This point is the critical problem on this paper.

*There are too many references. Because it is not a review
paper, Line 60 to Line 180 should be condensated.

Comments on the Quality of English Language

Nothing

Author Response

Reviewer1

Reviewers’ comment:

The conclusions are not consistent with the evidence and arguments presented and they don't address the main question posed. The authors mentioned that the first objective is coming up with a low-cost potable water monitoring system. However, nothing about them was shown and discussed. This point is the critical problem on this paper.
Response:

  • We greatly appreciate the feedback from the reviewer.
  • We have already linked the purpose argument (line 43) with the conclusion (line 191). To address the proposed system, we assembled our low-cost potable water monitoring system, which is shown in the manuscript Figure 3, is explained in lines 111-124, and is discussed in line 145-157.

Reviewers’ comment:

There are too many references. Because it is not a review paper, Line 60 to Line 180 should be condensated.

Response

  • We greatly appreciate the feedback from the reviewer.
  • We have eliminated some references and made the related works’ section concise. There are two subsections in the related works’ section: about ecosystem water quality monitoring (line 58-88), and subsection for machine learning (line 90-105).

Reviewers’ comment:

The amount of data being shown is very small. Therefore, it's unclear what the motivation of the research is and what conclusions were derived.

Response

  • We greatly appreciate the feedback from the reviewer.
  • We have already increased the sample quantity data from 2440 samples (figure 1) to become 2837 samples (figure 2) with self-assembled low-cost sensors correspondence with NB-IoT module for potable water monitoring in terms of cost adjustment and flexibility (shown in manuscript Figure 3 and is explained in line 112-124), even though the drinking water packages available on the marketplace. The new data collection has been reprocessed and then normalized. From the data collection, we compared the prediction models to determine drinkable or not drinkable water. We found the performance of the DT approach was fits the data.

Figure1. First data gathering

Figure2. Data normalization for Second data gathering

Reviewer 2 Report

Comments and Suggestions for Authors

Although the topic is interesting, the manuscript is not acceptable as the present form. All abbreviations should be spelled out when it first appears. Also, the citation is not complete in the manuscript (e.g., Line 23 ??). The reviewer suggest resubmitting the manuscript after revising the manuscript thoroughly. 

Reviewer 3 Report

Comments and Suggestions for Authors

The etitled paper "An IoT Real-Time Potable Water Quality Monitoring and Prediction Model based on Cloud Computing Architecture" investigates the use of IoT and Machine learning for drinking water monitoring. In this work authors present recents solutions then compare between them. Then, they proposed and discussed their presented work. The proposition is significan,  however, we recommand this changes:

**  All abbreviations should only be used after their first definition (write each abbreviation in it full time when it is used for the first time; repeat that in the abstract and in the main text) example: SDGs in the first line of the abstract.

 Don't repeat to write  Internet of Things (IoT)  in the other paragraphs (lines 66 -67 -116)

** Check English mistakes

** Check the refrences in line 23

** it seems that literature review is not very detailled, it is better to make a Table of related research.

**  add a sub-section where you discuss some related works (for water monitoring / prediction)

** give more details about your proposed framework

** Any details are giving about the features used by the MLP model in the section 3 

Author Response

Reviewers’ comment:

The entitled paper "An IoT Real-Time Potable Water Quality Monitoring and Prediction Model based on Cloud Computing Architecture" investigates the use of IoT and Machine learning for drinking water monitoring. In this work authors present recents solutions then compare between them. Then, they proposed and discussed their presented work. The proposition is significant, however, we recommend these changes:

**  All abbreviations should only be used after their first definition (write each abbreviation in it full time when it is used for the first time; repeat that in the abstract and in the main text) example: SDGs in the first line of the abstract.

 Don't repeat to write Internet of Things (IoT) in the other paragraphs (lines 66 -67 -116)

Response

  • We greatly appreciate the feedback from the reviewer.
  • The authors have already revised each abbreviation and not rewritten it. For instance definition Internet of Things (IoT) in line 4, the abbrevation in line 29, line 31, etc.

Reviewers’ comment:

** Check English mistakes

Response

  • We greatly appreciate the feedback from the reviewer.
  • We have already revised and checked the English mistakes.

Reviewers’ comment:

** Check the references in line 23

Response

  • We greatly appreciate the feedback from the reviewer.
  • The authors have already checked and completed the correct citation in the main text (line 23).

Reviewers’ comment:

** it seems that literature review is not very detailled, it is better to make a Table of related research.

Response

  • We greatly appreciate the feedback from the reviewer.
  • We have made a table of related research in Table 1 subsection IoT ecosystem of water quality monitoring, and Table 2, subsection prediction with machine learning model.

Reviewers’ comment:

**  add a sub-section where you discuss some related works (for water monitoring / prediction)

Response

  • We greatly appreciate the feedback from the reviewer.
  • We have made subsection IoT ecosystem of water quality monitoring, and subsection prediction with the machine learning model.

Reviewers’ comment:

** give more details about your proposed framework

Response

  • We greatly appreciate the feedback from the reviewer.
  • We have already detailed our proposed framework in manuscript figure 3 (line 108-125), and discussed in line 145-157.

Reviewers’ comment:

** Any details are giving about the features used by the MLP model in the section 3 

Response:

  • We thank the reviewer for pointing this out.
  • We have already detailed our proposed framework in manuscript section 3 (for example line 130-138).
